# Dapagliflozin Restores Impaired Autophagy and Suppresses Inflammation in High Glucose-Treated HK-2 Cells

**DOI:** 10.3390/cells10061457

**Published:** 2021-06-10

**Authors:** Jing Xu, Munehiro Kitada, Yoshio Ogura, Haijie Liu, Daisuke Koya

**Affiliations:** 1Department of Diabetology and Endocrinology, Kanazawa Medical University, Uchinada, Kahoku 920-0293, Ishikawa, Japan; xujing552417@hotmail.com (J.X.); namu1192@kanazawa-med.ac.jp (Y.O.); haijiecmu@126.com (H.L.); koya0516@kanazawa-med.ac.jp (D.K.); 2Division of Anticipatory Molecular Food Science and Technology, Medical Research Institute, Kanazawa Medical University, Uchinada, Kahoku 920-0293, Ishikawa, Japan

**Keywords:** sodium-glucose cotransporter 2, autophagy, inflammation, diabetic kidney disease

## Abstract

Sodium-glucose cotransporter2 (SGLT2) inhibitors have a reno-protective effect in diabetic kidney disease. However, the detailed mechanism remains unclear. In this study, human proximal tubular cells (HK-2) were cultured in 5 mM glucose and 25 mM mannitol (control), 30 mM glucose (high glucose: HG), or HG and SGLT2 inhibitor, dapagliflozin-containing medium for 48 h. The autophagic flux was decreased, accompanied by the increased phosphorylation of S6 kinase ribosomal protein (p-S6RP) and the reduced phosphorylation of AMP-activated kinase (p-AMPK) expression in a HG condition. Compared to those of the control, dapagliflozin and SGLT2 knockdown ameliorated the HG-induced alterations of p-S6RP, p-AMPK, and autophagic flux. In addition, HG increased the nuclear translocation of nuclear factor-κB p65 (NF-κB) p65 and the cytoplasmic nucleotide-binding oligomerization domain-like receptor 3 (NLRP3), mature interleukin-1β (IL-1β), IL-6, and tumor necrosis factorα (TNFα) expression. Dapagliflozin, SGLT2 knockdown, and NF-κB p65 knockdown reduced the extent of these HG-induced inflammatory alterations. The inhibitory effect of dapagliflozin on the increase in the HG-induced nuclear translocation of NF-κB p65 was abrogated by knocking down AMPK. These data indicated that in diabetic renal proximal tubular cells, dapagliflozin ameliorates: (1) HG-induced autophagic flux reduction, via increased AMPK activity and mTOR suppression; and (2) inflammatory alterations due to NF-κB pathway suppression.

## 1. Introduction

Diabetic kidney disease (DKD) is one of the main chronic complications of diabetes and the leading cause of end-stage renal disease (ESRD) [1]. Despite various therapeutic options for glucose control, many diabetic patients remain at high risk for development and progression of DKD. Therefore, there is an urgent need for more effective interventions for preventing the development and progression of DKD.

Sodium-glucose cotransporter 2 (SGLT2) is localized in the proximal tubule and is critical for 80–90% of the glucose filtered under physiological conditions [2]. SGLT2 inhibitors are effective antidiabetic drugs that attenuate hyperglycemia by suppressing renal glucose reabsorption. The results of large-scale clinical studies have shown that SGLT2 inhibitors improve hyperglycemia independently of insulin [3,4] and present renoprotective and cardioprotective effects in diabetic patients [5,6,7]. Although dapagliflozin treatment has led to remarkable renal outcomes [5,8], the relevant mechanisms have still not been fully elucidated.

Alterations in nutrient-sensing signaling pathways, including the AMP-activated kinase (AMPK) pathway, are involved in the pathogenesis of DKD through autophagy impairment and inflammation [9,10]. AMP-activated protein kinase (AMPK) is an energy sensor that responds to energy stress by restoring adenosine triphosphate (ATP) to maintain metabolic balance. Activation of AMPK is mainly required for the phosphorylation of the α subunit of Thr172 [11]. Autophagy is a process through which cellular proteins and organelles are degraded, and its activation is triggered by nutrient deficiency and suppressed by excess nutrition [12]. Therefore, nutrient-sensing pathways, including the AMPK and mechanistic target of rapamycin (mTOR) pathways, are the main regulators of autophagy. A previous study demonstrated that DKD is characterized by marked suppression of AMPK expression and activation of the mTOR pathway, leading to a decrease in autophagic flux in renal tubules [13]. Multiple studies have illustrated that SGLT2 inhibitors phosphorylated AMPK and activated autophagy, thereby exerting protective effects in diabetes and DKD [14,15,16].

Inflammation is also intimately involved in the pathogenesis of diabetes and DKD [10,17]. Nuclear factor-kappa B (NF-κB) represents a family of transcription factors that regulate multiple proinflammatory molecules, including interleukin (IL) and tumor necrosis factor-alpha (TNFα) [18]. NF-κB p65 (also known as RELA) is the most extensively studied subunit that contains transcriptional activation domains [18,19]. NF-κB p65 can shuttle from the cytoplasm to the nucleus in response to cell stimulation and is closely related to inflammatory activation in DKD [20]. In addition to proinflammatory cytokines, NF-κB can activate inflammasomes, such as the nucleotide binding and oligomerization domain-like receptor family pyrin domain-containing 3 (NLRP3) complex. NLRP3 inflammasome activation involves two signals: the NF-κB translocation-dependent expression of NLRP3 and precursor IL-1β [21,22], and the formation of mature IL-1β, as induced by the NLRP3 inflammasome [22]. These inflammatory cascade processes are crucial for the pathogenesis of DKD [10,23,24]. AMPK can suppress the activation of the NF-κB family and participate in posttranscriptional modifications of NF-κB [25]. Other studies have shown that SGLT2 inhibition suppresses inflammatory cytokine expression [21], reduces inflammasome activation [26,27], and protected the heart from inflammation via AMPK activation in diabetic mice [28]. However, it remains unclear whether the renal protective effect of SGLT2 inhibitors is exerted through the restoration of autophagy, and/or the attenuation of inflammation through nutrient-sensing signaling pathways.

Here, we hypothesized that the autophagy dysregulation and inflammation mediated by high glucose-induced alteration of nutrient-sensing pathways in cultured human renal proximal tubular cells is attenuated by the SGLT2 inhibitor dapagliflozin.

## 2. Materials and Methods

### 2.1. Cell Culture and Treatment

Human kidney proximal tubular cells (HK-2 cells, ATCC CRL-2190^TM^) were purchased from the American Type Culture Collection (ATCC, Manassas, VA, USA) and cultured in keratinocyte-SFM (K-SFM, Thermo Fisher Scientific, San Jose, CA, USA). When 70–80% confluent, the cells were cultured with high glucose stimulation in 30 mM glucose-containing medium (HG) for 48 h, while cells cultured in 5 mM glucose-containing medium supplemented with 25 mM mannitol were used as controls (Cont). The SGLT2 inhibitor dapagliflozin (DAPA, AstraZeneca, Cambridge, UK) was used at a final concentration of 10–100 μM for 48 h.

### 2.2. Transfection of Small Interfering RNAs (siRNAs)

HK-2 cells were cultured in K-SFM in six-well plates, and when 70–80% confluent, the cells were transfected with siRNA targeting SGLT2 (#289404, 200 nM), AMPK (#143192, 200 nM), RELA (#109424, 100 nM), NLRP3 (#107575, 200 nM) (Table 1), or a negative control siRNA (AM4635, siCON) with Lipofectamine 2000 (#2209775, Invitrogen, Carlsbad, CA, USA). The cells were incubated for 6 h, and then the medium was replaced with fresh K-SFM with or without HG (30 mM) for 48 h. All siRNAs were purchased from Thermo Fisher Scientific (San Jose, CA, USA).

### 2.3. Glucose Uptake

Glucose uptake was defined as the uptake of 2-deoxyglucose (2-DG) by HK-2 cells using a glucose uptake fluorometric assay kit (K666-100, BioVision, Milpitas, CA, USA) according to the manufacturer’s instructions. Generally, HK-2 cells were cultured in 96-well plates in keratinocyte SFM. When 50–60% confluent, the cells were treated with or without dapagliflozin (DAPA, 20 μM) for 48 h or cells were transfected with control siCON or siSGLT2 (200 nM) for 6 h. The cells were incubated in 0.5% bovine serum albumin (BSA)/phosphate-buffered saline (PBS) with or without 2-DG for 1 h. The relative fluorescence units (RFU) were measured at Ex/Em = 544/590 nm using a Fluoroskan Microplate Fluorometer (#5210470, Thermo Fisher Scientific, San Jose, CA, USA). The uptake of 2-DG in all samples was calculated by the 2-DG-6-phosphate (2-DG6P) standard curve and the RFU of samples.

### 2.4. Autophagic Flux Assay

HK-2 cells were treated with control (Cont), HG (30 mM), or HG (30 mM) with dapagliflozin (DAPA, 20 μM). After 48 h, chloroquine (CQ, 50 μM, C6628, Sigma-Aldrich, St. Louis, MO, USA) or vehicle was added, and the cells were incubated for 1 h before the collection of samples. Autophagic flux was calculated by the ratio of LC3-II in the sample treated with CQ and the sample treated without CQ. The ratio of LC3-II in the sample cultured in HG (30 mM) and rapamycin (0.4 μg/mL, 48 h, R0395, Sigma Aldrich, St. Louis, MO, USA) was used as a positive control for mTOR inhibition and autophagy activation. The samples were collected and analyzed by western blot analysis.

### 2.5. Protein Extraction and Western Blot Analysis

Total proteins were harvested by radioimmunoprecipitation lysis buffer with phenylmethanesulfonylfluoride, sodium orthovanadate, and a protease inhibitor cocktail (Santa Cruz Biotechnology, Dallas, TX, USA). Nuclear and cytoplasmic proteins were harvested using a nuclear extraction kit (ab113474, Abcam, Cambridge, UK) according to the manufacturer’s instructions. Concentrations of proteins were measured using a BCA protein assay kit (#23225, Pierce, Rockford, IL, USA) following the manufacturer’s instructions. Protein lysates were denatured in 4× Laemmli sample buffer (#161-0747, Bio-Rad, Hercules, CA, USA) at 100 °C for 5 min.

Equal amounts of protein lysates were separated on sodium dodecyl sulfate-polyacrylamide gel and transferred onto a polyvinylidene fluoride membrane by using the semidry method. After blocking with Tris-buffered saline containing 0.05% Tween 20 and 5% nonfat dry milk, the membranes were incubated overnight with primary antibodies targeting SGLT2, p-AMPKα (Thr172), AMPKα, LC3, p-S6RP (Ser235/236), S6RP, NF-κB p65, NLRP3, IL-1β, IL-6, TNFα, histone H3, and β-actin (Table 2) at 4 °C. The membranes were washed three times with Tris-buffered saline containing 0.05% Tween 20 and then incubated with HRP-conjugated secondary antibodies (#7074S, #7076S, 1:5000, Cell Signaling Technology, Danver, MA, USA) for 1 h at room temperature. After washing three times with Tris-buffered saline containing 0.05% Tween 20, the blots were developed with an enhanced chemiluminescence detection system (#32106, Pierce Biotechnology, Rockford, IL, USA) and visualized using an Image-Quant LAS 400 camera system (GE Healthcare Life Sciences, Uppsala, Sweden).

### 2.6. Immunofluorescence

HK-2 cells were cultured in K-SFM serum-free medium in eight-well plates. When the cells were 70–80% confluent, the medium was changed and the following treatments were administered before 48 h of incubation: control (Cont), HG (30 mM), HG (30 mM) with dapagliflozin (DAPA, 20 μM), or HG (30 mM) with rapamycin (rapa 0.4 μg/mL). Alternatively, cells were transfected with siSGLT2, as explained above. The cells were fixed with 4% paraformaldehyde for 30 min, permeabilized with 0.5% Triton X-100 for 15 min, blocked in 2% BSA/PBS for 30 min, and then incubated overnight with primary antibodies against IL-6 or p-S6RP (Table 2) at 4 °C. After being washed with PBS, the cells were incubated with secondary antibodies conjugated with Alexa Fluor 647 (Jackson ImmunoResearch, West Grove, PA, USA) for 30 min. After the cells were washed with PBS, the nuclei were counterstained with 4′,6-diamidino-2-phenylindole (DAPI), and the cells were observed under a fluorescence microscope (BZ-X700, Keyence, Osaka, Japan).

### 2.7. ELISA

HK-2 cells were cultured in K-SFM serum-free medium in six-well plates. When the cells were 70–80% confluent, cells were divided into 3 groups: control (Cont), HG (30 mM), and HG (30 mM) with dapagliflozin (DAPA, 20 μM). After 48 h, supernatants of cultured medium were collected. Levels of pro-inflammatory cytokines in the cultured medium were detected by using human ELISA kits targeting IL-1β (DLB50, R&D, Minneapolis, MN, USA), IL-6 (D6050, R&D, Minneapolis, MN, USA), and TNFα (DTA00D, R&D, Minneapolis, MN, USA), according to the manufacturer’s instructions. The optical densities (O.D.) of samples were measured and corrected by subtracting the readings at 570 nm from the readings at 450 nm using an Emax Plus Microplate Reader (Molecular Devices, San Jose, CA, USA). The concentrations of pro-inflammatory cytokines in cultured medium were calculated by the standard curve and the O.D. of samples.

### 2.8. Cytotoxicity Detection

To detect the cytotoxicity of dapagliflozin, HK-2 cells were treated with different concentrations of dapaliflozin (0–100 μM) under HG condition. After 48 h, the supernatants of the culture medium were collected. The activity of lactate dehydrogenase (LDH) in the culture medium that was released from damaged cells was determined using a Cytotoxicity Detection Kit (#04744926001, Roche, Mannheim, Germany) according to the manufacturer’s instructions. The O.D. of the sample was measured and corrected by subtracting the readings at 650 nm and the reading of cell-free culture medium from the readings at 492 nm using an Emax Plus Microplate Reader (Molecular Devices, San Jose, CA, USA).

### 2.9. Statistical Analysis

GraphPad Prism software (ver. 7.0f; La Jolla, CA, USA) was used for statistical analysis. The experimental results represent the mean ± standard deviation (SD) values of three independent duplicate experiments, and the significance of the differences between three or more groups was evaluated using one-way ANOVA followed by Tukey’s multiple comparison test. A value of *p* < 0.05 was considered statistically significant.

## 3. Results

### 3.1. Dapagliflozin and SGLT2 Inhibition Suppressed Glucose Uptake by HK-2 Cells

SGLT2 is critical for glucose absorption in the human proximal tubule. In HK-2 cells, HG induced the expression of SGLT2, and both dapagliflozin and SGLT2 knockdown (efficiency 41.5%) significantly suppressed SGTL2 expression and glucose uptake (Figure 1).

### 3.2. Dapagliflozin Suppressed the HG-Induced Reduction in Autophagic Flux through AMPK Activation

To determine the optimal concentration of dapagliflozin, we examined the dose effect of dapagliflozin in HG treated-HK-2 cells on cytotoxicity and AMPK activation. Compared with the Cont, HG (30 mM) did not increase cell death, but significantly suppressed the expression of p-AMPK. Dapagliflozin (10–100 μM) restored p-AMPK expression in a dose-dependent manner (Figure 2A,B). However, compared to the HG without dapagliflozin group, high concentrations of dapagliflozin (more than 50 μM) showed significant cytotoxicity in the high glucose condition (Appendix A). Then, we evaluated the AMPK-autophagy pathway in the treatment with dapagliflozin (20 μM) for 24 h and 48 h, respectively. The 24-h treatment with dapagliflozin (20 μM) ameliorated the decrease in p-AMPK and suppressed the increase in p-S6RP in high-glucose treated HK-2 cells. However, there were no significant changes in autophagic flux (Appendix A). If the duration of dapagliflozin was extended to 48 h, HG-impaired autophagic flux could be restored by dapagliflozin treatment (Figure 2C,D). On the other hand, dapagliflozin (20 μM) significantly ameliorated the decrease in p-AMPK, the increase in p-S6RP (a downstream protein of mTOR (Figure 3A,B), and the decrease in autophagic flux due to HG in the 48-h treatment (Figure 3E,F). To investigate the mechanism of dapagliflozin action on autophagic flux, AMPK siRNA was used to abrogate the activation of AMPK. The autophagic flux induced by dapagliflozin was abrogated by AMPK knockdown, which was accompanied by an increase in p-S6RP level (Figure 3A,B,E,F). Rapamycin, an inhibitor of the mTOR pathway, also restored the increase in p-S6RP expression (Figure 3C,D,G) and the reduction in autophagic flux caused by HG (Figure 3E,F). These results indicate that the inhibitory effect of dapagliflozin on the reduction in autophagic flux by HG may be exerted through the activation of AMPK and the decrease in mTOR activity.

### 3.3. SGLT2 Inhibition Suppressed HG-Induced Inflammatory Alterations

Compared to those treated with Cont, the cells treated with HG showed increased SGLT2 expression and decreased p-AMPK expression, as well as inflammatory alterations, including increased protein levels of IL-6 (Appendix A), TNFα, and mature IL-1β (Figure 4A,B,D,E) and secreted IL-6, TNFα, and IL-1β in cultured medium (Appendix A). Immunofluorescence staining also showed that, compared with those treated with Cont, the cells treated with HG induced the expression of IL-6. However, both dapagliflozin treatment (Figure 4A–C, Appendix A) and SGLT2 knockdown (Figure 4D–F, Appendix A) attenuated all of the inflammatory alterations induced by HG.

### 3.4. Dapagliflozin Suppressed Inflammatory Alterations by Inhibiting HG-Induced NF-κB Pathway Activation through the Activation of AMPK

To explore the mechanisms underlying the anti-inflammatory mechanism of SGLT2 inhibition, nuclear and cytoplasmic protein levels were measured after dapagliflozin treatment or SGLT2 knockdown. Compared to the effect of the Cont treatment, HG increased the nuclear translocation of NF-κB (p65), which was accompanied by increases in cytoplasmic NLRP3, mature IL-1β, and TNFα expression in the cytoplasmic fraction (Figure 5A–D). However, both dapagliflozin and SGLT2 knockdown significantly reduced the extent of the inflammatory alterations (Figure 5A–D). These results indicated that the anti-inflammatory effects of SGLT2 inhibition may contribute to the suppression of NF-κB p65 translocation in HG-treated HK-2 cells. To confirm our hypothesis, RELA siRNA (a small interfering RNA that selectively knocks down the NF-κB p65 subunit) was transfected into cells to block the aggregation of nuclear NF-κB p65 induced by HG. NF-κB p65 (RELA) knockdown (efficiency 86.2%) significantly decreased NLRP3, mature IL-1β, and TNFα expression (Figure 6A,B), and NLRP3 knockdown (efficiency 39.0%) suppressed the HG-induced increase in mature IL-1β expression (Figure 6C,D, Appendix A). These results indicate that dapagliflozin may reduce HG-induced inflammasome activation and increase TNFα expression through inhibition of NF-κB p65 nuclear translocation.

To investigate the effect of AMPK on inflammation, AMPK siRNA was transfected into cells to attenuate the effect of dapagliflozin. The inhibitory effect of dapagliflozin on the HG-induced increase in nuclear translocation of NF-κB p65 was abrogated by knocking down AMPK, which was accompanied by inflammatory alteration such as elevated NLRP3, mature IL-1β, IL-6, and TNFα expression (Figure 7A,B, Appendix A). These results indicated that the effect of dapagliflozin on inflammation may be dependent on the suppression of nuclear NF-κB p65 via AMPK activation.

## 4. Discussion

Tubular injury is one of the characteristics of DKD [10]. SGLT2 is critical for glucose reabsorption by proximal tubular cells [2]. HK-2 cells are well-differentiated proximal tubular cells which keep the basal functions of proximal tubular epithelium such as glucose reabsorption [29]. Previous studies showed that the expression of SGLT2 is high and stable from passage 6 to passage 12 and significantly decreased at passage 18 in HK-2 cells [30,31]. Moreover, although immune cells are the main source of pro-inflammatory cytokines in kidney, studies have confirmed that HG-induced pro-inflammatory cytokines in proximal tubular cells are also involved in tubular injury under a diabetic condition [20,32]. Therefore, HK-2 cells are suitable for exploring potential mechanisms of DKD.

Previous studies demonstrated that both SGLT2 knockdown [33] and SLGT2 inhibitors [34] significantly decreased glucose uptake by human proximal tubular cells. Our results (Figure 1) were consistent with the previous studies. AMPK is extremely sensitive to changes in energy status in response to cellular ATP consumption and the ratio of AMP or ADP/ATP [11,35]. Therefore, activation of AMPK is a key regulator of multiple nutritional and metabolic pathways, including gluconeogenesis, glycolysis, lipogenesis, protein synthesis, and autophagy pathways [11]. DKD is considered a state of overnutrition, with a marked inhibition of AMPK phosphorylation due to a decreased ratio of AMP/ATP or ADP/ATP [14,34,36]. Multiple studies have demonstrated that SGLT2 inhibitors including canagliflozin, empagliflozin, luseogliflozin, and ipragliflozin can activate p-AMPK [14,16,34,37,38]. A previous study showed that the SGLT2 inhibitor canagliflozin, not dapagliflozin or empagliflozin, had a dose-dependent effect (0–100 μM) on AMPK activation in HEK-293 cells [14]. In our study, dapagliflozin showed an effect equivalent to that of canagliflozin in inducing AMPK activity at concentrations from 10 to 100 μM in HK-2 cells. Monitoring the cytotoxicity of high concentrations of dapagliflozin is necessary in cell experiments. Our cytotoxicity detection showed that more than 50 µM of dapagliflozin may increase cell death (Appendix A), so the minimum effective concentration (20 µM) was safe and effective for further cell experiments in our study.

Autophagy impairment in renal cells is related to the pathogenesis of DKD [39]. Inhibition of AMPK caused by HG leads to the activation of the mTOR pathway, which contributes to renal hypertrophy and injury [40,41]. Our previous studies demonstrated that the suppression of AMPK and activation of the mTOR pathway contributed to HG-induced autophagy impairment under DKD conditions [42,43,44]. However, a recent study indicated that autophagy activity has the opposite manifestation in type 1 and type 2 diabetes mellitus (T1DM and T2DM) [45]. In a previous in vitro study, HG (treatment for 6–12 h, imitating T1DM) directly induced autophagic flux due to decreased lysosomal activity, while HG suppressed autophagic flux only in the presence of insulin (imitating T2DM) via mTOR activation [45]. In our study, HG induced p-S6RP increases with or without insulin (100 nM), indicating HG can directly activate the mTOR pathway, which was consistent with results from previous research [40,41,42,43]. Although compared with HG, the activation of the mTOR pathway in the presence of insulin was more obvious, there was no difference in the suppression of p-AMPK and autophagic flux (Appendix A). The difference between our study and the previous study may be based on the incubation time of cells with HG medium (24–48 h vs. 6–12 h). Previous studies demonstrated that SGLT2 inhibitors enhance autophagy, including increases in the microtubule-associated protein 1A/1B-light chain 3-II (LC3-II) and decreased p62 via the AMPK/mTOR pathway [15,16,46]. In our study, with the suppression of p-AMPK via HG or HG plus insulin, autophagic flux was significantly suppressed. Dapagliflozin treatment can restore HG-impaired p-AMPK and autophgic flux in the presence and absence of insulin (Appendix A). These data indicated that dapagliflozin can restore impaired autophagic flux in both a T1DM and T2DM condition in vitro. In addition, AMPK knockdown via siRNA induced p-S6RP expression and suppressed autophagic flux. Moreover, the mTOR inhibitor rapamycin activated autophagic flux by suppressing p-S6RP expression. All this evidence indicated that dapagliflozin restored the HG-induced autophagy impairment previously induced through via the AMPK/mTOR pathway, which is consistent with the reports of previous research [15,16]. There was an interesting phenomenon in our experiment. Compared with other groups, basal LC3-II was significantly increased in the AMPK-knockdown group. However, this change did not indicate that autophagy was activated, as chloroquine did not increase the expression of LC3-II in the AMPK knockdown group. We speculated that the increase in basal LC3-II induced by AMPK knockdown may be due to the failure of autophagosome clearance.

Inflammation is another significant feature of DKD [47]. Hyperglycemia activates the production of multiple proinflammatory cytokines (e.g., IL-1, IL-6, and TNFα) and leads to the recruitment of inflammatory cells to the kidney [48,49]. Previous studies demonstrated that SGLT2 inhibitors decreased the accumulation of nuclear NF-κB p65 in db/db mice [50], high-fat diet-fed Wistar rats [51], and HG-treated HK-2 cells [52] and then suppressed downstream proteins, including TNFα and IL-1β [51]. All these studies indicated that the translocation of the NF-κB p65 subunit is crucial for activating inflammation in DKD. On the other hand, the release of IL-1β is controlled by NLRP3 inflammasome activation [22,23,53]. NLRP3 activation may be involved in multiple inflammatory or stress pathways, including NF-κB signaling pathways [23]. Lipopolysaccharide (LPS)-induced NF-κB can bind to the NLRP3 promoter to directly regulate its expression in human macrophages [21], which indicates that NLRP3 is a downstream gene of NF-κB. Multiple studies have shown that SGLT2 inhibition suppresses the NLRP3 inflammasome in different diabetic animal models [26,27,54,55,56]. Dapagliflozin suppressed the activation of the NLRP3 inflammasome in diabetic myocardial tissues [26] and reduced the production of IL-1β in aortic and liver tissues [54,55]. AMPK can suppress inflammatory signaling [25] and NLRP3 activation [57,58]. Other studies demonstrated that AMPK activation may inhibit the protein modification of NF-κB p65, including its phosphorylation [58,59,60]. As a transcriptional regulator of a variety of proinflammatory cytokines, the nuclear translocation of NF-κB p65 is an important step for the activation of proinflammatory cytokines. In the present study, we found that the increased translocation of NF-κB p65 from the cytoplasm to the nucleus was critical for the inflammation caused by HG and that AMPK activation suppresses these changes. Our findings provide another potential mechanism of dapagliflozin in the inflammation induced via AMPK/NF-κB p65/NLRP3 signaling in human kidney proximal tubular cells. Our study illustrated the possible mechanisms of the renoprotection of dapagliflozin in the treatment of DKD and suggested other potential benefits of dapagliflozin beyond glucose control. However, as this was only an in vitro study, our results need to be further verified in rodent models of T1DM and T2DM in the future.

## 5. Conclusions

In diabetic renal proximal tubular cells, HG-induced decreased AMPK activity is involved in (1) mTOR activation and autophagic flux reduction and (2) inflammatory alterations (inflammasome activation and increased TNFα expression) due to NF-κB (p65) pathway activation (Figure 8). Dapagliflozin ameliorates the alterations caused by HG.

## Figures and Tables

**Figure 1 cells-10-01457-f001:**
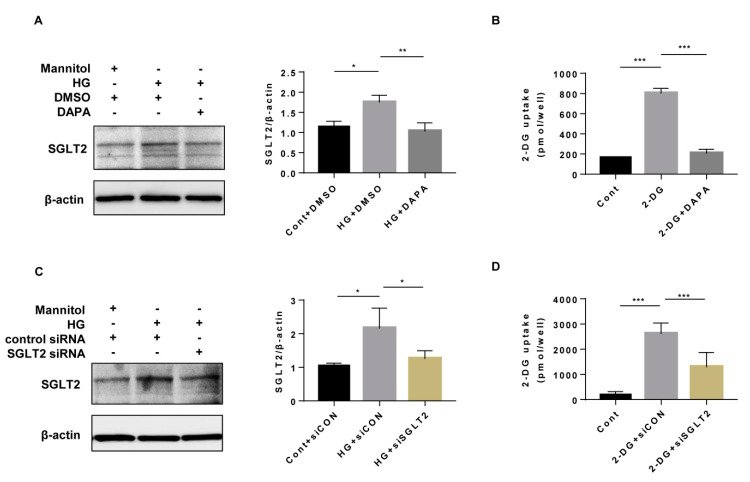
Dapagliflozin and SGLT2 inhibition suppressed glucose uptake in HK-2 cells. (**A**) HK-2 cells were incubated with Cont or HG (30 mM) medium and treated with or without dapagliflozin (20 µM) for 48 h. SGLT2 was assessed by western blot densitometric evaluation (n = 3) of (**A**). (**B**) Glucose uptake analysis of 2-DG by dapagliflozin-treated HK2 cells (n = 6). (**C**) HK-2 cells were transfected with control siRNA or SGLT2 siRNA (200 nM) for 6 h and then, the medium was replaced with fresh medium with or without HG (30 mM) for 48 h. The SGLT2 level was assessed by western blotting (n = 3). (**D**) Glucose uptake analysis of 2-DG by SGLT2-knockdown HK-2 cells (n = 6). All data represent the means ± standard deviation (SD). * *p* < 0.05 vs. the indicated group, ** *p* < 0.01 vs. the indicated group, *** *p* < 0.001 vs. the indicated group. Cont, control; HG, high glucose; DAPA, dapagliflozin; siCON, control siRNA; and siSGLT2, SGLT2 siRNA.

**Figure 2 cells-10-01457-f002:**
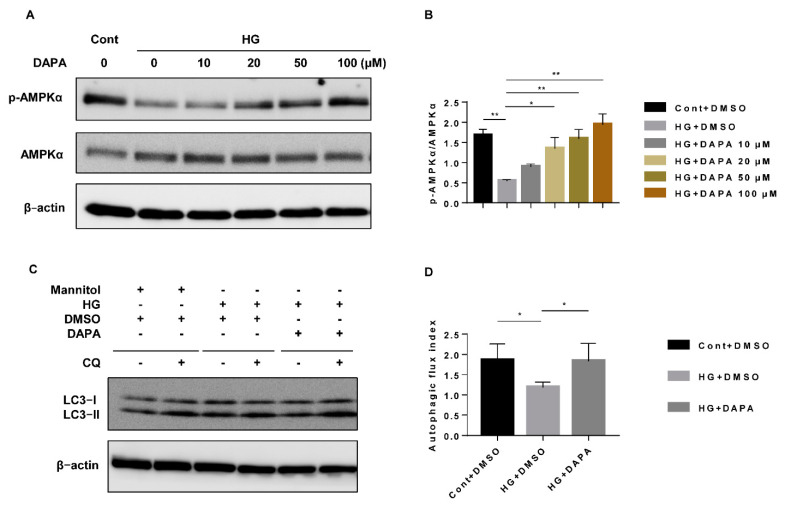
Dapagliflozin increased the activity of AMPK and suppressed the reduction in autophagic flux due to HG. (**A**) HK-2 cells were treated with dapagliflozin (0–100 μM, for 48 h) under HG conditions, and p-AMPK was assessed by western blotting. (**B**) Densitometric evaluation of (**A**) (n = 3). (**C**) HK-2 cells were treated with Cont, HG (30 mM), HG (30 mM) with dapagliflozin (20 μM) for 48 h, and chloroquine (50 μM) or vehicle for 1 h. Autophagic flux was determined with the LC3-II ratio in the presence and absence of chloroquine. (**D**) Densitometric evaluation of (**C**) (n = 5). All data represent the means ± standard deviation (SD). * *p* < 0.05 vs. the indicated group, ** *p* < 0.01 vs. the indicated group. Cont, control; HG, high glucose; DAPA, dapagliflozin; and CQ, chloroquine.

**Figure 3 cells-10-01457-f003:**
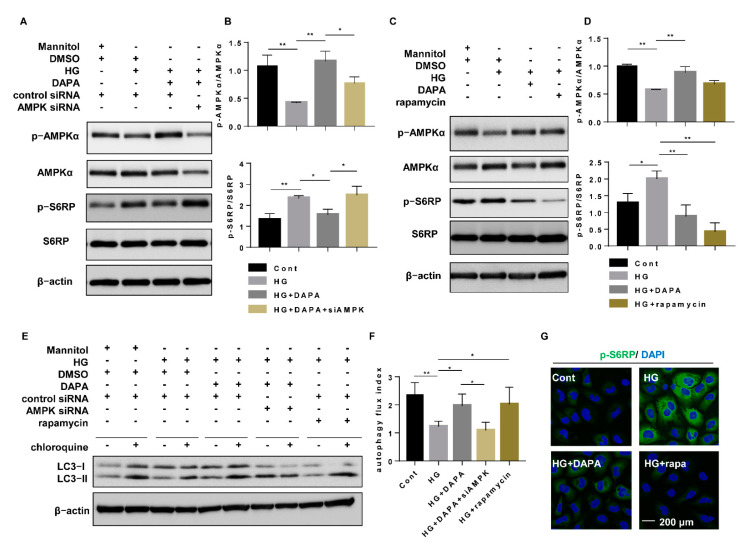
Dapagliflozin restored HG-impaired autophagy via the AMPK/mTOR pathway. (**A**) HK-2 cells were transfected with AMPK siRNA (200 nM) or control siRNA for 6 h and then incubated with Cont, HG (30 mM), or HG (30 mM) with dapagliflozin (20 µM) for 48 h. The p-AMPK, AMPK, p-S6RP, and S6RP levels were assessed by western blotting. (**B**) Densitometric evaluation of (**A**) (n = 3). (**C**) HK-2 cells were incubated with Cont, HG (30 mM), HG (30 mM) with dapagliflozin (20 µM), or HG (30 mM) with rapamycin (0.4 µg/mL) for 48 h, and the p-AMPK, AMPK, p-S6RP, and S6RP levels were assessed by western blotting. (**D**) Densitometric evaluation of (**C**) (n = 3). (**E**) Quantitative autophagic flux is represented by western blots of LC3-II inHK-2 cells cultured with Cont, HG, dapagliflozin, dapagliflozin with siAMPK, and rapamycin in the presence and absence of chloroquine (50 μM for 1 h). (**F**) Densitometric evaluation of (**E**) (n = 5). All data represent means ± standard deviation (SD). (**G**) Immunofluorescence of p-S6RP in HK-2 cells cultured with Cont, HG, dapagliflozin, dapagliflozin with siAMPK, and rapamycin. * *p* < 0.05 vs. the indicated group, ** *p* < 0.01 vs. the indicated group. Cont, control; HG, high glucose; DAPA, dapagliflozin; siCON, control siRNA; siAMPK, AMPK siRNA; and CQ, chloroquine.

**Figure 4 cells-10-01457-f004:**
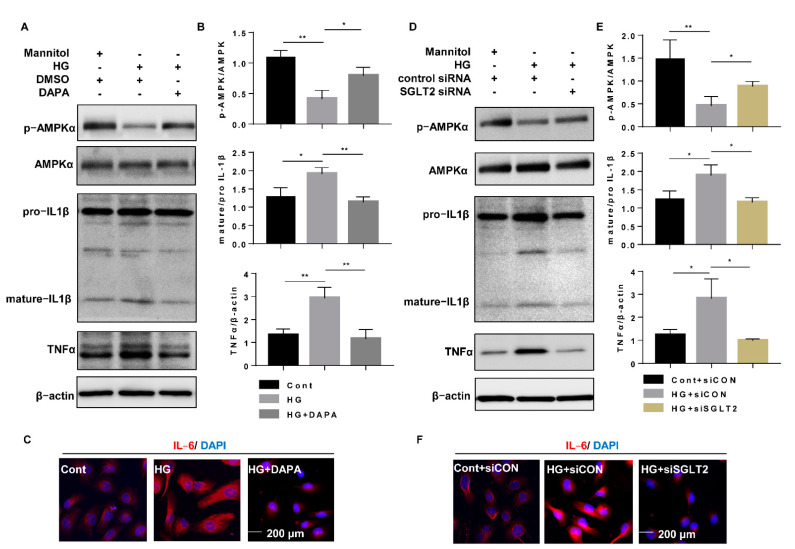
SGLT2 inhibition suppressed HG-induced inflammatory alterations. (**A**) HK-2 cells were incubated with Cont, HG (30 mM), or HG (30 mM) dapagliflozin (20 µM) for 48 h, and p-AMPK, AMPK, IL-1β, and TNFα were assessed by western blotting. (**B**) Densitometric evaluation of (**A**) (n = 3). (**C**) Immunofluorescence of IL-6 in HK-2 cells cultured with Cont, HG (30 mM), or HG (30 mM) dapagliflozin (20 µM) for 48 h. (**D**) HK-2 cells were transfected with SGLT2 siRNA (200 nM) or control siRNA for 6 h and then incubated with Cont or HG (30 mM) medium for 48 h. P-AMPK, AMPK, IL-1β, and TNFα were assessed by western blotting. (**E**) Densitometric evaluation of (**D**) (n = 3). (**F**) Immunofluorescence of IL-6 in HK-2 cells transfected with SGLT2 siRNA (200 nM) or control siRNA for 6 h and then treated with Cont or HG (30 mM) medium for 48 h. All data represent the mean ± standard deviation (SD). * *p* < 0.05 vs. the indicated group, ** *p* < 0.01 vs. the indicated group (n = 3). Cont, control; HG, high glucose; DAPA, dapagliflozin; siCON, control siRNA; siSGLT2; and SGLT2 siRNA.

**Figure 5 cells-10-01457-f005:**
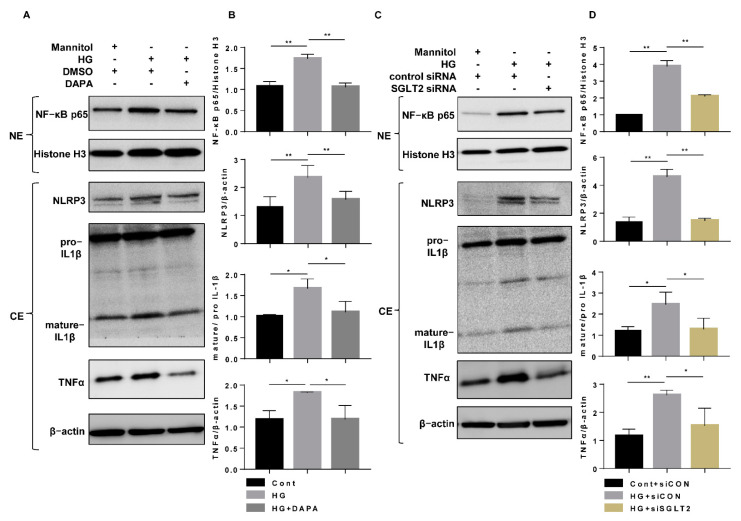
SGLT2 inhibition suppressed HG-induced nuclear localization of NF-κB p65. (**A**) HK-2 cells were incubated with Cont, HG (30 mM), or HG (30 mM) and dapagliflozin (20 µM) for 48 h, and the levels of NF-κB p65 in nucleoprotein extracts and NLRP3, IL-1β, TNFα in cytoplasmic protein extracts were assessed by western blotting. (**B**) Densitometric evaluation of (**A**) (n = 3). (**C**) HK-2 cells were transfected with SGLT2 siRNA (200 nM) or control siRNA for 6 h and then incubated with Cont or HG (30 mM) medium for 48 h. The levels of NF-κB p65 in nucleoprotein extracts and NLRP3, IL-1β, and TNFα in cytoplasmic protein extracts were assessed by western blotting. (**D**) Densitometric evaluation of (**C**) (n = 3). All data represent means ± standard deviation (SD). * *p* < 0.05 vs. the indicated group, ** *p* < 0.01 vs. the indicated group (n = 3). Cont, control; HG, high glucose; DAPA, dapagliflozin; siCON, control siRNA; siSGLT2, SGLT2 siRNA; NE, nucleoprotein extracts; and CE, cytoplasmic protein extracts.

**Figure 6 cells-10-01457-f006:**
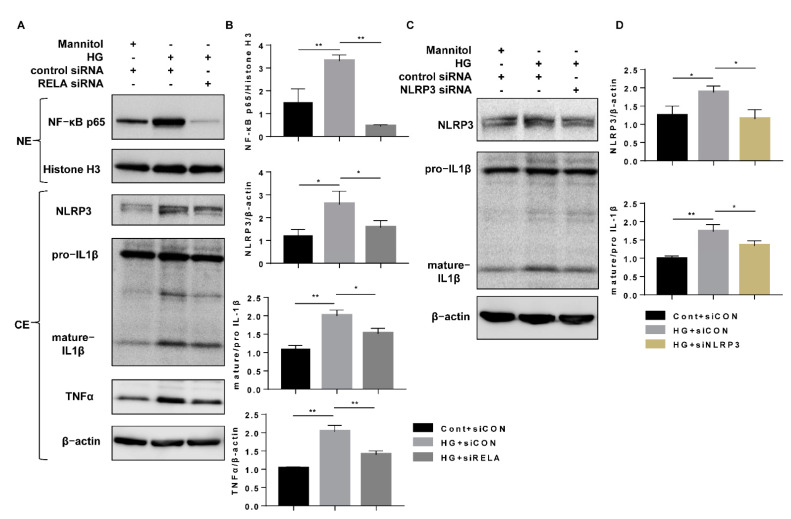
HG-induced inflammatory alteration via nuclear NF-κB p65. (**A**) HK-2 cells were transfected with RELA siRNA (100 nM) or control siRNA for 6 h and then incubated in Cont or HG (30 mM) medium for 48 h. The levels of NF-κB p65 in nucleoprotein extracts and NLRP3, IL-1β, and TNFα in cytoplasmic protein extracts were assessed by western blotting. (**B**) Densitometric evaluation of (**A**) (n = 3). (**C**) HK-2 cells were transfected with NLRP3 siRNA (200 nM) or control siRNA for 6 h and then incubated with Cont or HG (30 mM) medium for 48 h. The levels of NLRP3 and IL-1β were assessed by western blotting. (**D**) Densitometric evaluation of (**C**) (n = 3). All data represent means ± standard deviation (SD). * *p* < 0.05 vs. the indicated group, ** *p* < 0.01 vs. the indicated group. Cont, control; HG, high glucose; siCON, control siRNA; siRELA, RELA siRNA; siNLRP3, NLRP3 siRNA; NE, nucleoprotein extracts; and CE, cytoplasmic protein extracts.

**Figure 7 cells-10-01457-f007:**
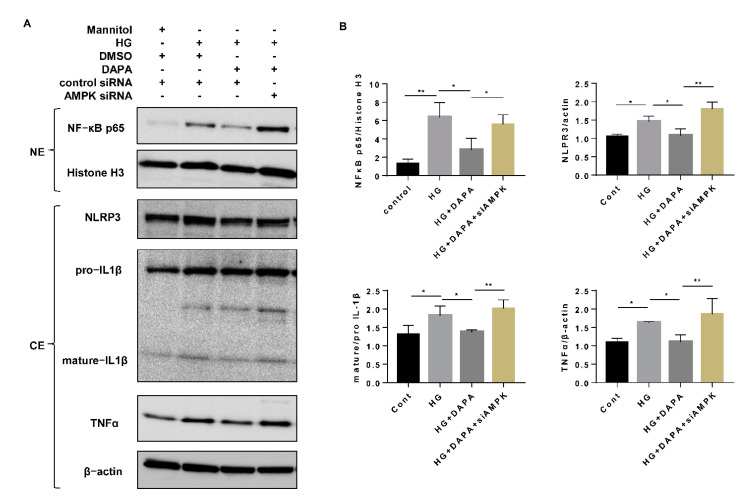
Dapagliflozin suppressed HG-induced nuclear NF-κB p65 by activating AMPK. (**A**) HK-2 cells were transfected with AMPK siRNA (200 nM) or control siRNA for 6 h and then incubated with Cont, HG (30 mM), or HG (30 mM) with dapagliflozin (20 µM) for 48 h. The levels of NF-κB p65 in nucleoprotein extracts and NLRP3, IL-1β, and TNFα in cytoplasmic protein extracts were assessed by western blotting. (**B**) Densitometric evaluation of (**A**) (n = 3). All data represent means ± standard deviation (SD). * *p* < 0.05 vs. the indicated group, ** *p* < 0.01 vs. the indicated group (n = 3). Cont, control; HG, high glucose; siCON, control siRNA; siAMPK, AMPK siRNA; NE, nucleoprotein extracts; CE, and cytoplasmic protein extracts.

**Figure 8 cells-10-01457-f008:**
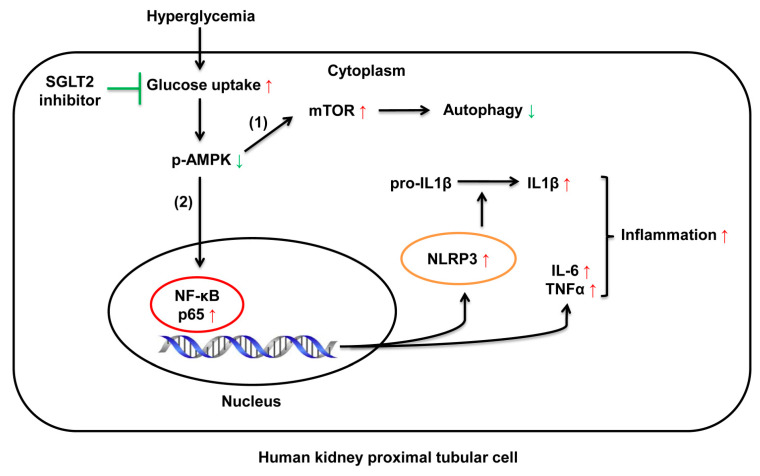
The mechanism of SGLT2 inhibition in diabetic renal proximal tubular cells. High glucose-induced decrease in AMPK activity is involved in (1) mTOR activation and autophagic flux reduction and (2) inflammatory alterations (inflammasome activation and increased TNFα and IL-6 expression) due to NF-κB (p65) pathway activation. Dapagliflozin attenuates the alterations caused by high glucose.

**Table 1 cells-10-01457-t001:** Sequences of the small interfering RNAs used in this study.

Target	5′-3′ Sense	5′-3′ Antisense
SGLT2	GUCAUUGCUGCAUAUUUCCTT	GGAAAUAUGCAGCAAUGACTA
AMPK	GGUAGAUAUAUGGAGCAGUTT	ACUGCUCCAUAUAUCUACCTC
RELA	GGCUAUAACUCGCCUAGUGTT	CACUAGGCGAGUUAUAGCCTC
NLRP3	GGUGUUGGAAUUAGACAACTT	GUUGUCUAAUUCCAACACCTG

**Table 2 cells-10-01457-t002:** Antibodies used for western blotting and immunofluorescence.

Antigen	Species	Dilution (WB/IF ^1^)	Reference
SGLT2	Rabbit polyclonal	1:200 (WB)	GeneTex, GTX59872
p-AMPKα (Thr172)	Rabbit monoclonal	1:1000 (WB)	Cell Signaling Technology, #2535
AMPKα	Rabbit polyclonal	1:1000 (WB)	Cell Signaling Technology, #2532
LC3	Rabbit polyclonal	1:1000 (WB)	Cell Signaling Technology, #4108
p-S6RP (Ser235/236)	Rabbit polyclonal	1:1000 (WB) 1:100 (IF)	Cell Signaling Technology, #2211
S6RP	Rabbit monoclonal	1:1000 (WB)	Cell Signaling Technology, #2217
NF-κB p65	Rabbit monoclonal	1:1000 (WB)	Cell Signaling Technology, #4764
NLRP3	Rabbit monoclonal	1:500 (WB)	Cell Signaling Technology, #13158
IL-1β	Rabbit polyclonal	1:500 (WB)	Abcam, ab9722
IL-6	Mouse monoclonal	1:1000 (WB) 1:200 (IF)	Abcam, ab9234
TNFα	Rabbit polyclonal	1:1000 (WB)	Abcam, ab66579
Histone H3	Rabbit polyclonal	1:1000 (WB)	Cell Signaling Technology, #9715
β-actin	Mouse monoclonal	1:10,000 (WB)	Sigma Aldrich, A5316

^1^ WB, western blotting; IF, immunofluorescence.

## Data Availability

The data presented in this study are available on reasonable request from the corresponding author.

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
