# Peer review of "Dapagliflozin Restores Impaired Autophagy and Suppresses Inflammation in High Glucose-Treated HK-2 Cells"

_cells, 2021, doi:10.3390/cells10061457_

Round 1

Reviewer 1 Report

   The authors demonstrated that SGLT2-mediated glucose entry attenuates AMPK activity leading to stimulation of inflammatory pathways and downregulation of autophagy under a hyperglycemic condition in cultured renal proximal tubular cells. The findings in this study may help readers understand molecular mechanisms underlying mitigation of the development of renal proximal tubular damage by SGLT2 inhibitors in diabetes mellitus as the authors aimed. However, there are still several concerns which must be appropriately addressed.

Major comments

1) According to the authors’ statement, one of the overall goals in this study is delineating mechanisms underlying mitigation of the development of renal proximal tubular damage by SGLT2 inhibitors in diabetes mellitus. SGLT2 inhibitors have been approved to treat patients with type 2 diabetic mellitus (T2DM) involving hyperinsulinemia. However, the experiments were conducted in the absence of insulin and clinical relevance of the findings is not clear. Therefore, the elucidated mechanisms in this cell study should be demonstrated in a T2DM animal model and/or clinical subjects. Alternatively, confirm the major findings in HK-2 cells supplemented with insulin.

2) It is very surprising that SGLT2, RelA and NLRP3 siRNAs did not suppress target protein levels to significantly lower than each control (Fig. 1C, Fig. 6B and 6D). Thus, add a group “Normal glucose (=Mannitol) + siRNA” for the knockdown experiments, respectively.

3) Phosphorylation levels of AMPK and S6RP were evaluated at 48 hours of treatments. In general, activation of intracellular signal transducers can be observed even within one hour. Thus, show temporal changes of phosphorylation levels of these factors from the early time point.

4) Pro-inflammatory cytokines measured in this study can be secreted to the culture medium. Therefore, determine the cytokine levels in the medium. In addition, explain why IL-6 levels were shown only by immunostaining although the other cytokines were detected by western blots.

5) Indeed, proximal tubular cells express pro-inflammatory cytokines. However, the major sources of the cytokines may be immune cells accumulated in the kidneys during the development of diabetic nephropathy. Demonstrate contribution of the HK-2 cell-derived cytokines to cell damage under the diabetic condition.

Minor comments

1) Some supportive data may help to provide firm evidence. Detect p62 levels in addition to demonstrated LC3B conversion. Moreover, evaluate NF-kB activity by an additional method such as phosphorylation of p65 and/or p50, degradation of IkB, or DNA binding of p65/p50.

2) Add S6RP to Fig. 8.

Author Response

Response to Reviewer 1 Comments

We thank reviewers and editors for proving us fruitful comments and for the chance to revise our manuscript. We are also thankful to the reviewer’s efforts for spending their valuable time in reading the manuscript. These suggestive comments from reviewers are very convincing. Authors are providing answers to all the comments by point to point raised by reviewers’ comments.   

Reviewer 1

 The authors demonstrated that SGLT2-mediated glucose entry attenuates AMPK activity leading to stimulation of inflammatory pathways and downregulation of autophagy under a hyperglycemic condition in cultured renal proximal tubular cells. The findings in this study may help readers understand molecular mechanisms underlying mitigation of the development of renal proximal tubular damage by SGLT2 inhibitors in diabetes mellitus as the authors aimed. However, there are still several concerns which must be appropriately addressed.

Major comments

1) According to the authors’ statement, one of the overall goals in this study is delineating mechanisms underlying mitigation of the development of renal proximal tubular damage by SGLT2 inhibitors in diabetes mellitus. SGLT2 inhibitors have been approved to treat patients with type 2 diabetic mellitus (T2DM) involving hyperinsulinemia. However, the experiments were conducted in the absence of insulin and clinical relevance of the findings is not clear. Therefore, the elucidated mechanisms in this cell study should be demonstrated in a T2DM animal model and/or clinical subjects. Alternatively, confirm the major findings in HK-2 cells supplemented with insulin.

 ----- We thank reviewer’s positive evaluation and constructive comments on our manuscript. This is one of limitations in our experiment. We summarized it in discussion (Line 377-380).

2) It is very surprising that SGLT2, RelA and NLRP3 siRNAs did not suppress target protein levels to significantly lower than each control (Fig. 1C, Fig. 6B and 6D). Thus, add a group “Normal glucose (=Mannitol) + siRNA” for the knockdown experiments, respectively.

 ----- We thank reviewer’s comments. This is an important point. The expression of SGLT2, RelA and NLRP3 is relatively low in the control group. Our purpose is to observe the changes under the high-sugar environment, so we did not check the knockdown efficiency in normal glucose group. The efficiency of knocking out target genes in high glucose group are 41.5% (P=0.046), 86.2% (p=0.0002) and 39.0% (P=0.016), respectively (Line 161, Line 250 and Line 251). The differences are statistically significant. We believe that these siRNAs are effective. We also did the experiment of NLRP3 knockdown in the control group and the high glucose group at the same time. As was shown in Figure S3, the expression of NLRP3 was very weak in the control group, and there was no significant difference between control and the knockdown of the control group, so we did not set knockout of the control group in the experiment.

3) Phosphorylation levels of AMPK and S6RP were evaluated at 48 hours of treatments. In general, activation of intracellular signal transducers can be observed even within one hour. Thus, show temporal changes of phosphorylation levels of these factors from the early time point.

 ----- We thank reviewer’s evaluation and constructive comments. HG-induced chronic change is more important in our model. So we evaluated phosphorylation levels of AMPK, S6RP and autophagic flux at 24h (which was not shown in our manuscript) and 48h. As is shown in Figure S1, 24h treatment of dapagliflozin (20 μM) ameliorated the decrease in p-AMPK and suppressed the increase in p-S6RP in high glucose-treated HK-2 cells. However, there was no obvious change in autophagic flux. Therefore, we speculate that although the AMPK pathway can be activated in a short time, the feedback of downstream such as autophagy may take longer (Line 177-182).

4) Pro-inflammatory cytokines measured in this study can be secreted to the culture medium. Therefore, determine the cytokine levels in the medium. In addition, explain why IL-6 levels were shown only by immunostaining although the other cytokines were detected by western blots.

 ----- We thank reviewer’s constructive comments and we are sorry for the negligence on IL-6. In fact, we also checked the protein levels of IL-6 via western blot but did not put it in the original manuscript. As is shown in Figure S2, both dapagliflozin treatment and SGLT2 knockdown can suppress high glucose-induced IL-6 expression. In our experiment, nearly all the target proteins were checked by western blot, so we want to use different experimental methods to verify the conclusions. So we used immunofluorescence to detect IL-6, which can also confirm the inflammatory changes.

5) Indeed, proximal tubular cells express pro-inflammatory cytokines. However, the major sources of the cytokines may be immune cells accumulated in the kidneys during the development of diabetic nephropathy. Demonstrate contribution of the HK-2 cell-derived cytokines to cell damage under the diabetic condition.

 ----- We thank reviewer’s constructive comments. Previous studies have demonstrated that HK-2 cell-derived cytokines induced by high glucose contributed to tubular injury in the pathogenesis of DKD (Line 305-308 and reference 20, 32).

Minor comments

1) Some supportive data may help to provide firm evidence. Detect p62 levels in addition to demonstrated LC3B conversion. Moreover, evaluate NF-kB activity by an additional method such as phosphorylation of p65 and/or p50, degradation of IkB, or DNA binding of p65/p50.

----- We thank reviewer’s constructive comments. According to the “Guidelines for the use and interpretation of assays for monitoring autophagy (4th edition)”, multiple methods can be used to measure autophagic flux. Chloroquine (CQ) is also recommended: “Chloroquine inhibits the fusion between autophagosomes and lysosomes, preventing the maturation of autophagosomes into autolysosomes, and blocking a late step of autophagy” (PMID: 33634751). As a transcriptional regulator of multiple proinflammatory cytokines, the nuclear translocation of NF-κB p65 is an important step for the activation of inflammation. Therefore, we believe that the detection of autophagic flux by CQ and the nuclear NF-κB p65 translocation are sufficient to explain the state of autophagy and inflammation in our experiment.

Reviewer 2 Report

This is an interesting in vitro investigation of the efficacy of SGLUT-2 inhibitor Dapagliflozin in restoring impaired autophagy and control of inflammation in high glucose-treated HK-2 cells.

The work is well designed, and I have no concerns regarding the laboratory protocol which appears to be rigorous and accurately conceived.

However, given the completely experimental nature of the study, its scientific soundness and overall take-home message might benefit from some short hints on the transferability on the clinical practice.

Some points might be better detailed in the discussion section also in view of the recent literature, that is giving a growing emphasis to the “hot-topic” of the role of the renoprotective effects of SGLUT-2 inhibitors.

  • At lines 298-301, the authors make a comparison of dapagliflozin with canagliflozin to discuss the differences in terms of dose-dependent effect on AMPK activation in cultured cells. It would be interesting to include some more information on other glifozins, namely empagliflozin, ertugliflozin, ipragliflozin, in view of their growing application in renal patients as they have been proven to be safe and to confer an effective protection against renal and cardiovascular events.
  • To give a more complete view on the role of Dapagliflozin and the clinical applicability of these in vitro data, it might also be of interest to discuss briefly the implications effects of SGLT2 on electrolyte balance, which is an important point as mineral metabolism perturbations induced by SGLT2i administration and the related risk of bone fractures (Cianciolo G et al, JBMR Plus. 2019 doi: 10.1002/jbm4.10242).

In summary, this is an interesting, well-written and well-designed work and I have no concerns regarding the presentation of methods and results. It that might benefit from some clarifications in the discussion sections in light of the abovementioned points and the suggested recent references to be added.

Author Response

Response to Reviewer 2 Comments

We thank reviewers and editors for proving us fruitful comments and for the chance to revise our manuscript. We are also thankful to the reviewer’s efforts for spending their valuable time in reading the manuscript. These suggestive comments from reviewers are very convincing. Authors are providing answers to all the comments by point to point raised by reviewers’ comments.   

Reviewer 2

This is an interesting in vitro investigation of the efficacy of SGLUT-2 inhibitor Dapagliflozin in restoring impaired autophagy and control of inflammation in high glucose-treated HK-2 cells.

The work is well designed, and I have no concerns regarding the laboratory protocol which appears to be rigorous and accurately conceived.

However, given the completely experimental nature of the study, its scientific soundness and overall take-home message might benefit from some short hints on the transferability on the clinical practice.

Some points might be better detailed in the discussion section also in view of the recent literature, that is giving a growing emphasis to the “hot-topic” of the role of the renoprotective effects of SGLUT-2 inhibitors.

1) At lines 298-301, the authors make a comparison of dapagliflozin with canagliflozin to discuss the differences in terms of dose-dependent effect on AMPK activation in cultured cells. It would be interesting to include some more information on other glifozins, namely empagliflozin, ertugliflozin, ipragliflozin, in view of their growing application in renal patients as they have been proven to be safe and to confer an effective protection against renal and cardiovascular events.

----- We thank reviewer’s constructive comments. In fact, reference 14 evaluated the activation of AMPK by three different SGLT2 inhibitors including canagliflozin, dapagliflozin and empagliflozin (which we did not mention in our manuscript) at different concentrations and found only canagliflozin had a dose-denpendent effect on AMPK activation. We also added some other evidence on the activation of AMPK by empagliflozin and irpagliflozin in discussion (Line 316-317). By now, there is no research on the effect of ertugliflozin on AMPK.

2) To give a more complete view on the role of Dapagliflozin and the clinical applicability of these in vitro data, it might also be of interest to discuss briefly the implications effects of SGLT2 on electrolyte balance, which is an important point as mineral metabolism perturbations induced by SGLT2i administration and the related risk of bone fractures (Cianciolo G et al, JBMR Plus. 2019 doi: 10.1002/jbm4.10242).

----- We thank reviewer’s constructive comments. We discussed briefly the relationship between the electrolyte balance, the risk of bone fractures and SGLT2 inhibitors in discussion (Line 316-319).

In summary, this is an interesting, well-written and well-designed work and I have no concerns regarding the presentation of methods and results. It might benefit from some clarifications in the discussion sections in light of the abovementioned points and the suggested recent references to be added.

----- We thank reviewer’s evaluation and constructive comments.

Reviewer 3 Report

  1. Does this effect occur only in Dapagliflozin or can investigators elaborate into other SGLT2i?
  2. The hypothesis of the study should be explicitly stated
  3. The authors should discuss what the clinical utility of these findings could be  
  4. There should be more discussion of a possible mechanism
  5. Explain what sort of future study might take us closer to future development and a clinical utility from these findings?

Author Response

Response to Reviewer 3 Comments

We thank reviewers and editors for proving us fruitful comments and for the chance to revise our manuscript. We are also thankful to the reviewer’s efforts for spending their valuable time in reading the manuscript. These suggestive comments from reviewers are very convincing. Authors are providing answers to all the comments by point to point raised by reviewers’ comments.   

Reviewer 3

  • Does this effect occur only in Dapagliflozin or can investigators elaborate into other SGLT2i?

 ----- We thank reviewer’s comments. Other SGLT2 inhibitors including canagliflozin, empagliflozin, luseogliflozin and irpagliflozin can also activate AMPK (Line 316-319).

  • The hypothesis of the study should be explicitly stated

 ----- We thank reviewer’s comments. The hypothesis of the study is that high glucose may lead to autophagy dysregulation and inflammation, and SGLT2 inhibitor dapagliflozin can restore high glucose-impaired autophagy and suppress high glucose-induced inflammation via AMPK activation.

  • The authors should discuss what the clinical utility of these findings could be

---- We thank reviewer’s comments. Our study illustrated the possible mechanisms of the renoprotection of dapagliflozin in the treatment of DKD. These results may suggest other benefits of dapagliflozin beyond glucose control.

4) There should be more discussion of a possible mechanism

----- We thank reviewer’s comments. Our study confirmed that the renoprotection of dapagliflozin under high glucose environment was mainly dependent on suppressing inflammation and restoring autophagy via AMPK activation. We added some new references and explanations in discussion (Line 301-308, Line 370-384).

5) Explain what sort of future study might take us closer to future development and a clinical utility from these findings?

----- We thank reviewer’s comments. This was only an in vitro study, so the results need to be further verified in rodent models of T1DM and T2DM in the future (Line 377-380).

Reviewer 4 Report

There are 2 major methodological problems with this study.

Confluent epithelial cells should be used to examine the renal tubular system if one is already working in such an artificial system as immortal HK2.
In addition, it was shown that HK-2 cells have only low SGLT2 expression compared to other cell lines (Kuang et al., https://www.medscimonit.com/abstract/index/idArt/902530). The authors should use a better system (primary cells, not HK-2). That should be discussed. 

On the other hand, the selected concentration range of dapagliflozin is absolutely unusually high. Why are you only working in concentrations above 10µM and not working in normal human plasma/serum concentrations (up to 500nM) or used in comparable in vitro studies (10nM to 500nM, but not more than 2 µM)?

Using such high dapagliflozin concentrations, the authors would also have to investigate the cytotoxicity at these high concentrations and discuss the meaning of such high dapagliflozin concentrations in their study.

Author Response

Response to Reviewer 4 Comments

We thank reviewers and editors for proving us fruitful comments and for the chance to revise our manuscript. We are also thankful to the reviewer’s efforts for spending their valuable time in reading the manuscript. These suggestive comments from reviewers are very convincing. Authors are providing answers to all the comments by point to point raised by reviewers’ comments.   

Reviewer 4
There are 2 major methodological problems with this study.

1) Confluent epithelial cells should be used to examine the renal tubular system if one is already working in such an artificial system as immortal HK2. In addition, it was shown that HK-2 cells have only low SGLT2 expression compared to other cell lines (Kuang et al., https://www.medscimonit.com/abstract/index/idArt/902530). The authors should use a better system (primary cells, not HK-2). That should be discussed. 

----- We thank reviewer’s evaluation and constructive comments. Indeed, primary cells are the best model for in vitro study. HK-2 cells are well-differentiated proximal tubular cells and retain function of proximal tubular epithelium such as Na+ dependent/phlorizin sensitive glucose transport (reference 29). So HK-2 cells are widely used in the experiments of kidney disease, especially tubular injury. Previous studies also showed that SGLT2 is highly expressed and stable from passage 6 to passage 12, while significantly decreased at passage 18 in HK-2 cells (reference 30, 31). Based on these studies, HK-2 cells from passage 8 to passage 12 were used in our experiment (Line 302-308).

2) On the other hand, the selected concentration range of dapagliflozin is absolutely unusually high. Why are you only working in concentrations above 10µM and not working in normal human plasma/serum concentrations (up to 500nM) or used in comparable in vitro studies (10nM to 500nM, but not more than 2 µM)?

Using such high dapagliflozin concentrations, the authors would also have to investigate the cytotoxicity at these high concentrations and discuss the meaning of such high dapagliflozin concentrations in their study.

----- It is a great question and indeed crossed our mind as well. In previous studies, the concentration of dapagliflozin used in HK-2 cells ranged from 1 µM to 100 µM (reference 14, 52). One study evaluated the activation of AMPK by three different SGLT2 inhibitors including canagliflozin, dapagliflozin and empagliflozin at different concentrations (0-100 µM) in HEK293 cells (reference 14). Another study demonstrated that dapagliflozin inhibited inflammation in HK-2 cells at the concentration of 10 µM-100 µM (reference 52). Based on these research, we chose the concentration of 10-100 µM on AMPK activation. The results showed that 20 µM -100 µM dapagliflozin had a significant effect on AMPK activation. A recent study also showed that dapagliflozin over 10 µM may cause cytotoxicity (reference 31). In our study, we also observed cell death at 100 µM, so the minimum effective concentration in our experiment (20 µM) was chosen for further research. Indeed, this concentration also exceeded the in normal human plasma/serum concentrations. Further experiments are required for monitoring cytotoxicity of high concentrations of dapagliflozin (Line 381-384).

We replaced Figure 4 to 7 without changing the original images, because the mistakes of labeling in the figures (all the places labeled “LG” were replaced by “Cont”.)

Thank you for your time and consideration of our response to the critiques submitted by four reviewers. We will be happy to address any further questions that arise.

Round 2

Reviewer 1 Report

The authors have responded adequately to some of my comments and suggestions and improved the manuscript. However, they still need to provide key results itemized below to support the conclusions as requested in my original critiques. Alternatively, the authors should explain about any difficulties to conduct the requested in vitro experiments.

  1. I understood the limitations of this study added to the manuscript. However, many studies demonstrated that insulin prevents high glucose-induced renal cellular damages, suggesting enhanced glucose influx via SGLT2 may not contribute to diabetic cellular damage in insulin-treated cells (T2DM conditions). In addition, testing some major findings in the presence of insulin using the in vitro setting may not be an inevitable limitation. Accordingly, confirm the major findings in HK-2 cells supplemented with insulin.
  2. Even if proximal tubular cell-derived cytokines, rather than cytokines derived from immune cells, could be the dominant pathogenic factors in the cell damages, secreted cytokines induce the cell damages in autocrine or paracrine manners. Therefore, measurements of cytokine levels in the culture medium are required.
  3. In case there are any difficulties to perform the experiments listed above, describe the reasons as a response to the critiques (not in the manuscript).

Author Response

The authors have responded adequately to some of my comments and suggestions and improved the manuscript. However, they still need to provide key results itemized below to support the conclusions as requested in my original critiques. Alternatively, the authors should explain about any difficulties to conduct the requested in vitro experiments.

  1. I understood the limitations of this study added to the manuscript. However, many studies demonstrated that insulin prevents high glucose-induced renal cellular damages, suggesting enhanced glucose influx via SGLT2 may not contribute to diabetic cellular damage in insulin-treated cells (T2DM conditions). In addition, testing some major findings in the presence of insulin using the in vitro setting may not be an inevitable limitation. Accordingly, confirm the major findings in HK-2 cells supplemented with insulin.

----- We thank reviewer’s constructive comments on our manuscript. We checked the p-AMPK, p-S6RP and autophagic flux in insulin-treated HK-2 cells. Our data indicated that HG induced p-S6RP, suppressed p-AMPK, autophagic flux in the absence and presence of insulin and discussed these results in discussion part (Figure S4 and Line 359-371).

  1. Even if proximal tubular cell-derived cytokines, rather than cytokines derived from immune cells, could be the dominant pathogenic factors in the cell damages, secreted cytokines induce the cell damages in autocrine or paracrine manners. Therefore, measurements of cytokine levels in the culture medium are required.

----- We thank reviewer’s constructive comments. We checked secreted cytokines including IL-6, IL-1β and TNFα in cultured medium by ELISA and got the similar results as western blotting (Figure S5 A-C and Line 245-250).

In case there are any difficulties to perform the experiments listed above, describe the reasons as a response to the critiques (not in the manuscript).

----- We thank reviewer’s constructive comments. These comments helped us to better improve our experiments.

Thank you for your time and consideration of our response to the critiques submitted by four reviewers. We will be happy to address any further questions that arise.

Sincerely,

Munehiro Kitada, M.D., Ph.D

kitta@kanazawa-med.ac.jp

Reviewer 3 Report

all of my comments have been properly addressed. 

Author Response

Our manuscript was edited by "Springer Nature Authors Services".

Thank you for your time and consideration of our response to the critiques submitted by four reviewers. We will be happy to address any further questions that arise.

Reviewer 4 Report

As I stated in the first review: Using such high dapagliflozin concentrations, the authors would also have to investigate the cytotoxicity at these high concentrations.

A determination of the cytotoxicity is absolutely necessary and must be included. I think it is no problem to perform a cytotox assay with HK-2 cells  (in triplicate (at minimum) and as biological replicates, and with 6-7 concentrations of Dapa, up to 100µM).

Author Response

As I stated in the first review: Using such high dapagliflozin concentrations, the authors would also have to investigate the cytotoxicity at these high concentrations. A determination of the cytotoxicity is absolutely necessary and must be included. I think it is no problem to perform a cytotox assay with HK-2 cells (in triplicate (at minimum) and as biological replicates, and with 6-7 concentrations of Dapa, up to 100µM).

----- We thank reviewer’s constructive comments. Indeed, detecting cytotoxicity of dapagliflozin is crucial for our experiments. We detected the cytotoxicity of different concentrations of dapagliflozin (0, 1, 10, 20, 50, 100 µM for 48 h) according to the reviewer’s comments by using cytotoxicity detection kit based on LDH activity. The results showed that more than 50 µM dapagliflozin significantly increased cell death (Figure S5D). Combined with the results of dapagliflozin's effect on AMPK activation, we believe that 20 µM dapagliflozin is safe and effective for cell experiments (Line 196-202 and Line 347-350).

Thank you for your time and consideration of our response to the critiques submitted by four reviewers. We will be happy to address any further questions that arise.

Sincerely,

Munehiro Kitada, M.D., Ph.D

kitta@kanazawa-med.ac.jp

Round 3

Reviewer 1 Report

No further comments

Reviewer 4 Report

The authors have processed my points of criticism.